# Nutrition status and morbidity of Ethiopian children after recovery from severe acute malnutrition: Prospective matched cohort study

**Tsinuel Girma**[1]*, **Philip T. James**[2,3], **Alemseged Abdissa**[4,5], **Hanqi Luo**[2,6], **Yesufe Getu**[2,7], **Yilak Fantaye**[2,8], **Kate Sadler**[2], **Paluku Bahwere**[2,9]

**1** Department of Pediatrics and Child Health, Jimma University, Jimma, Ethiopia, **2** Valid International, Oxford, United Kingdom, **3** Emergency Nutrition Network, Kidlington, United Kingdom, **4** Department of Laboratory Sciences, Jimma University, Jimma, Ethiopia, **5** Armauer Hansen Research Institute, Addis Ababa, Ethiopia, **6** Department of Global Health, Rollins School of Public Health, Emory University, Atlanta, GA, United States of America, **7** Department of Physiology, Jimma University, Jimma, Ethiopia, **8** Alameda County Public Health Department, Oakland, CA, United States of America, **9** Center for Epidemiology, Biostatistics, and Clinical Research, School of Public Health, Free University of Brussels, Brussels, Belgium

* tsinuel@yahoo.com

**Data Availability Statement:** All relevant data are within the manuscript and its Supporting Information files.

## Abstract

After recovery, children with severe acute malnutrition (SAM) remain vulnerable to sub-optimal growth and malnutrition relapse. Although there is an increased interest in understanding these problems, data are scarce, and contextual factors can cause variability. We prospectively followed a cohort of Ethiopian children (215 post-SAM cases and 215 non-wasted controls), monthly for one year. The post-SAM cases were: age 6–59 months at admission into the community management of acute malnutrition (CMAM) program and being successfully discharged from CMAM (MUAC>11.0cm, weight gain of 20%, absence of oedema and clinically stable for two consecutive weeks). The controls were apparently healthy children from same village who had no history of an episode of AM and were matched 1:1 to a post-SAM child by age and sex. The primary outcomes were: cumulative incidence of acute malnutrition; growth trajectory; cumulative incidence of reported common morbidities, and cumulative proportion and incidence of deaths. The burden of common morbidities was higher among post-SAM than controls; post-SAM children had more frequent illness episodes (Incidence Rate Ratio of any illness 1.39, 95% CI: 1.14, 1.71; p<0.001). The prevalence of SAM was consistently higher among post-SAM cases than the control group, having a 14 times higher risk of developing SAM (Incidence Rate Ratio: 14.1; 95% CI: 3.5, 122.5; p<0.001). The divergence in weight and growth trajectory remained the same during the study period. Our results advocate for the design of post-discharge interventions that aim to prevent the reoccurrence of acute malnutrition, reduce morbidity and promote catch-up growth. Research is needed to define the appropriate package of post-discharge interventions.

**Funding:** TG,PB,KS,PJ.The funder was United States Agency for International Development (USAID) under Agreement No. AID-663-A-11-00017. The funders had no role in study design, data collection and analysis, decision to publish, or preparation of the manuscript.

**Competing interests:** The authors have declared that no competing interests exist.

## Introduction

The community management of acute malnutrition (CMAM) approach has increased the access and coverage of treatment for severe acute malnutrition (SAM) and moderate acute malnutrition (MAM) in most low- and middle-income countries (LMIC), including Ethiopia [1–4]. Remarkable results have been achieved with reports of recovery in children as high as 80% [5–8]. After nutritional therapy, however, recovered children remain at risk of relapse to MAM or SAM as they often return to the same living condition that does not support optimal growth. Although there is an increased interest in understanding relapse, data are scarce, and the published reports are difficult to compare as there is no standardized definition of relapse and universally agreed indicators [9].

Anthropometric criteria and the absence of nutritional oedema are currently used as proxies of physical and physiological recovery from SAM. The treatment ends when children have corrected their body mass deficit as measured by the weight-for-height index or mid-Upper Arm Circumference (MUAC) of $\geq$-2 Z-score or $\geq$12.5 cm, respectively, and/or they have no clinically detectable nutritional oedema [10]. Data are lacking to confirm if these children retain the weight and MUAC velocities that allowed the rapid correction of wasting and a return to their normal growth track. Information is also needed on the effect of the correction of wasting on the subsequent linear growth.

In children who have re-gained their body mass deficit, the complex immune dysfunctions associated with SAM are expected to improve [11, 12]. However, measuring the immune markers is methodologically challenging, and interpretation of the data generally is not straightforward [13]. Thus, the prevalence and incidence of common morbidities are often used as proxy measures of immune recovery.

The limited available evidence suggests that relapses are common and linear growth catch-up is insufficient [9, 14–17]. However, the reported relapse rates reach to 37% and vary across countries. Therefore, this study aimed to assess children's nutritional status and morbidity after recovery from severe acute malnutrition.

## Methods

### Study design

This prospective matched cohort study was conducted from the September 2013 to September 2015 in the rural population of Jimma Zone, Oromiya region in Southwest Ethiopia. Of the 18 districts (woredas) in the Jimma zone, three woredas (Dedo, Omonada, and Seka) were selected purposefully due to their high caseload of SAM and accessibility. We based our sample size estimation on the following assumptions: We projected an 8.5% difference between post-SAM and controls in cumulative incidence of acute malnutrition (AM), anticipating that over the one year follow up 13.1% of post-SAM children will have experienced an episode of SAM (95% confidence interval of ±2.5%: CI limit 7.5% to 12.5%).

This was based on the prevalence of SAM in Oromiya Region reported in the 2011 Ethiopian Demographic and Health Survey report [18]. With an assumed design effect of 1.5 and a loss to follow up of 20%, the required sample size was 474 (237 post-SAM and 237 controls) for a power of 80% and an alpha level of 5%. For the post-SAM cases, the inclusion criteria were: age 6–59 months at admission into the CMAM program and being successfully discharged from CMAM as per the national guideline criteria at the time (MUAC>11.0cm, weight gain of 20%, absence of oedema and clinically stable for two consecutive weeks).

Controls were eligible if they were apparently healthy with no history of an episode of AM and were matched 1:1 to a post-SAM child by age and sex. The mother/caretaker of the case

was asked to indicate the neighboring household having a child of the same sex and age. Study investigators followed up, and the child meeting the criteria closest to the case household was selected. The acceptable difference between case and control age was age ±3 months up to 24 months of age and ±6 months for the older children.

Mother/caregiver's consent for their children to participate and residence in the study catchment area for at least one year after enrolment were additional criteria for both groups. We excluded children with physical disability and any congenital disease that affected growth or prevented accurate anthropometric measurement (both groups), discharged directly from the inpatient nutrition rehabilitation unit for post-SAM, and presence of acute malnutrition (MUAC<12.0 cm or bilateral pitting pedal oedema) for the control group.

Both post-SAM and control groups were followed up concurrently at their homes monthly by trained data collectors over one year or until the participants died, decided to withdraw or move out of the area. Weight, length/height, and MUAC were measured according to WHO standards and in duplicate [19]. Length/height was measured to the nearest 0.1 cm using the UNICEF recommended portable wooden length/height board with an upright wooden base and movable headpiece. Children younger than 24 months were measured in supine position, and older children were measured while standing. Weight was measured to the nearest 0.1 kg using mother and child battery-powered SECA weighing scales (SECA 874, Hamburg, Germany). Younger children were weighed undressed while being held by their mothers and older children stepped on the scale. MUAC was measured at the midpoint of the left arm, using a non-stretch insertion tape to the nearest 0.1cm.

Morbidity data were collected using the two-week recall technique and are based on the caregiver's report of fever, diarrhoea, persistent cough and fast breathing during the two weeks prior to the home visit. Trained research nurses conducted voluntary HIV testing and counselling for all children of both groups.

Relapse of malnutrition was defined as follows: AM, MUAC <12.0 cm or pitting pedal oedema); moderate acute malnutrition (MAM), MUAC 11.0–12.0 cm and SAM, MUAC <11.0 cm or pitting bipedal oedema. The cut-offs values were based on the Ethiopian protocol that was used at the time of the study [20]. We did not restrict the definition of relapse to a fixed time interval after recovery from SAM, but included all relapse cases within our study period.

## Study outcomes

The primary outcomes of interest were: cumulative incidence of AM, MAM and SAM; growth trajectory (weight length/height, MUAC and related indices changes and trends); cumulative incidence of reported common morbidities, and cumulative proportion and incidence of deaths).

## Statistical analysis

Indices for weight-for-height (WHZ), length/height-for-age (HAZ), weight-for-age (WAZ), length/height-for-age difference (HAD) and weight-for-age difference (WAD) were calculated according to WHO (2006) growth standards (21). WHZ, HAZ and WAZ were calculated using the zscore06 command in Stata [21, 22]. HAD and WAD were calculated using the approach proposed by Leroy et al. [23]. HAD and WAD are the actual deficit when the child is compared to the median child of the same age and sex. We obtained a z-score by dividing this value by the corresponding standard deviation (SD). Because the SD is not a constant, this division introduces a mathematical artifact. It has been clearly demonstrated that HAD better captures the magnitude of the absolute deficit in children than HAZ and that HAZ may give a

false impression of growth catch up [23]. WHZ, HAZ and WAZ at enrolment were categorized into <-2 z-scores, ≥-2 to define absence or presence of wasting, stunting, and underweight, respectively. Both HAZ and HAD were used to characterize linear growth better. Based on the literature, accelerated linear growth (ALG) was diagnosed when HAZ increased by ≥0.67 between two assessments [23–26]. The covariant for outcome of wasting and morbidity was assessed by fitting mixed-effects models to include independent variables as covariates. A linear mixed-effects model was applied for the continuous outcome measures (rates of WHZ, WAZ, HAZ, BAZ and MUAC changes) and time to relapse of MAM and SAM. Time to malnutrition relapse was analyzed using the Cox proportional hazard models and Kaplan Meir survival analysis. All P values are two tailed, and statistical significance was set at p-value less than 0.05 with 95% CI.

### Ethical considerations

The ethical review board of Jimma University approved the study (reference RPGC/130/2013). Written informed consent was obtained from the caregiver of all participants. During the follow-up, children who developed SAM at any time were referred to the nearest health facility for appropriate dietary and medical management.

## Results

The initial sample size of 474 was not reached within the planned time of recruitment. However, the loss to follow-up was two-fold lower than anticipated, making the actual sample size of 430 (215 post-SAM cases and 215 non-wasted controls) sufficient to maintain the study power. Of the 430 enrolled children, follow-up data for analysis were available for 94.4% (203/215) post-SAM and 93.9% (202/215) controls (Fig 1).

As shown in Table 1, no difference was observed between post-SAM and control groups for matching criteria at baseline except that the post-SAM children were significantly lighter and shorter (p<0.001). Household characteristics were mostly comparable between the two group (S1 Table). All participants had an HIV-negative test result. Additionally, lost-to-follow cases and retained cases for both groups had comparable profiles.

In the post-SAM, the prevalence of MAM decreased from 7.5% to 5% by the end of the six-month follow-up (Fig 2). Meanwhile, the point prevalence of SAM (monthly) was consistently higher among post-SAM than the control group, with peaks at the third (3.6%) and seventh month (3.8%). In contrast, the control group had only two SAM cases at the eleventh month. Over the year, the post-SAM group had 14 times the risk of developing SAM than controls (Incidence Rate Ratio: 14.1; 95% CI: 3.5, 122.5; p<0.001) (Table 2). There were six deaths, all from post-SAM; six of them had AM (two with SAM). All deaths except one were disease-related, which was due to accident.

As Table 3 indicates, the post-SAM group were 1.4 times likely to have any of the common morbidities (cough, diarrhoea, fever or difficulty breathing) compared with the controls during the one-year follow-up period (Incidence Rate Ratio of any illness 1.39, 95% CI: 1.14, 1.71; p<0.001). Whereas, the risk for the individual morbidities was at least 1.7 times in the post-SAM compared to controls (p<0.001 for all individual morbidities).

The burden of common morbidities was higher (p<0.001) among post-SAM than controls (Fig 3); post-SAM children had more frequent illness episodes. On the other hand, the proportion of children who never reported the symptoms were lower for post-SAM children than non-wasted controls.

Table 4 shows anthropometric parameters at baseline and their changes over 12 months. Although the post-SAM group were discharged from CMAM as recovered, they were still

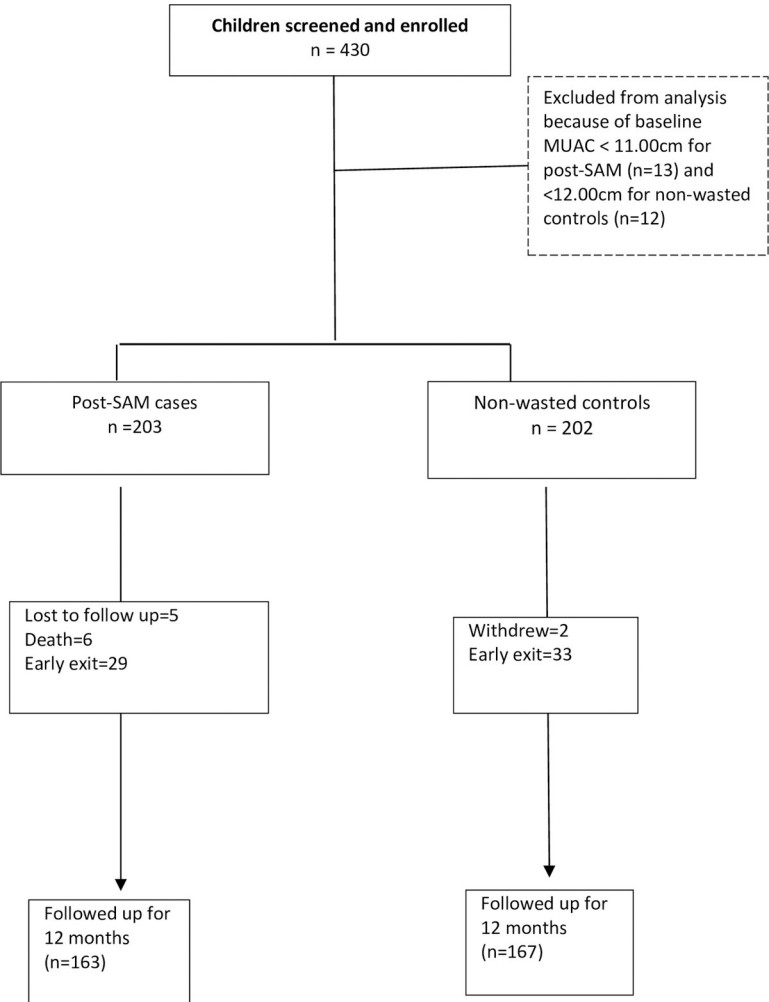

**Fig 1. Flow diagram showing enrolment and follow up of participants.**

significantly lighter, shorter, more wasted and more stunted than controls, both at baseline and 12 months later. The post-SAM group were less wasted at 12 months than at baseline but compared to controls they did not significantly reduce their deficit in weight, or MUAC by the 12th month follow up, except when WAZ is considered. At 12 months, both the cases and controls were more stunted than at baseline, with a more pronounced drop in linear growth in the controls. This translated into a significant reduction of the deficit in length/height only when HAZ was considered, but not when absolute length/height or HAD was used for comparison. The indices WAD and HAD showed that the widening of the deficit in weight and length/height was larger than depicted by the indices WAZ and HAZ and showed weight deficit widening even among post-SAM for whom WHZ and WAZ showed some catch up.

Fig 4 presents the overall trend of the anthropometric parameters during the 12 months of follow-up. The differences in weight, height and MUAC between post-SAM and controls did not reduce or widen during the follow up though WAZ and WHZ differences were slightly decreasing over time. Both HAZ and HAD curves showed that the post-SAM group did not experience catch-up linear growth compared to the control group and the age and sex corresponding WHO 2006 medians (S1 Fig). However, a quarter of post-SAM children [40/

**Table 1. Comparison of characteristics of the participants by study group and follow-up status.**

| | Follow up for 12 months | | | Follow up <12 months | | |
|---|---|---|---|---|---|---|
| | Post-SAM (n = 163) | Non-wasted control (n = 167) | p-value | Post-SAM (n = 41) | Non-wasted control (n = 37) | p-value |
| | n (%); median (IQR)[1] | n(%); median (IQR)[1] | | n (%); median (IQR)[1] | n(%); median (IQR)[1] | |
| Age (months) | 15 (11,31) | 15 (11,32) | 0.924 | 13 (11,24) | 13 (10,24) | 0.624 |
| Sex, female | 76 (45.5) | 76.(46.6) | 0.839 | 18 (43.9) | 18 (48.6) | 0.675 |
| BCG scar | 96 (58.9) | 112 (67.9) | 0.301 | 24 (58.5) | 29 (78,4) | 0.061 |
| Ever immunized | 133 (81.6) | 142 (85.0) | 0.402 | 30(73.2) | 30 (81.1) | 0.408 |
| Ever received vitamin A | 147 (90.2) | 156 (93.4) | 0.285 | 37 (90.2) | 32 (86.5) | 0.604 |
| Ever dewormed | 88 (54.0) | 79 (47.3) | 0.225 | 21 (51.2) | 18 (48.6) | 0.821 |
| Utilizes bed net, yes | 78 (47.8) | 91 (54.5) | 0.247 | 24 (58.5) | 27 (72.3) | 0.247 |
| MUAC[2] (cm) | 12.5 (12.0, 13.3) | 13.6 (13.0,14.4) | <0.001 | 12.2 (12.0, 12.9) | 13.5 (13.0,14.2) | <0.001 |
| Weight (kg) | 7.6 (6.4, 9.4) | 9.2 (8.2, 11.3) | <0.001 | 7.0 (6.4, 8.4) | 8.7 (7.8, 9.9) | <0.001 |
| Height (cm) | 70.3 (66.3, 76.3) | 74.6 (70.1, 85.4) | <0.001 | 68.4 (65.9, 73.0) | 74.0 (69.2, 79.8) | <0.001 |
| Z-score (WHO 2006) | | | | | | |
| Weight-for-age | -2.7 (-3.4, -2.0) | -1.2 (-1.8, -0.4) | <0.001 | -2.9 (-3.7, -1.8) | -0.9 (-2.0, -0.2) | <0.001 |
| Height-for-age | -3.4 (-4.4, -2.3) | -1.4 (-2.5, -0.7) | <0.001 | -3.4 (-5.3, -2.2) | -1.5 (-2.0, -0.1) | <0.001 |
| Weight-for-height | -1.1 (-2.0, -0.4) | -0.5 (-1.1, 0.3) | <0.001 | -1.5 (-2.8, -0.4) | -0.7 (-1.8, 0.3) | <0.001 |
| Absolute difference[3] | | | | | | |
| Weight-for-age (Kg) | -2.7 (-3.7, -2.0) | -1.2 (-2.2, -0.4) | <0.001 | -2.7 (-3.5, -1.9) | -1.0 (-1.6, -0.3) | <0.001 |
| Height-for-age (cm) | -8.9 (-13.0, -5.4) | -3.5 (-6.6, -1.2) | <0.001 | -8.6 (-13.0, -4.5) | -3.5 (-6.1, 0.0) | <0.001 |

156 = 25.6 (18.7; 32.6) %] had a HAZ increase of ≥0.67 by the 12th month follow-up, meeting the definition of ALG. Among post-SAM who were stunted at baseline, stunting reversal was observed in 14.4% [18/125 = 14.4 (8.1; 20.6) %]. The figure was 22.9% [11/148 = 22.9 (10.6; 35.2) %] among the controls who were stunted at baseline. Both HAZ and HAD had an upward trend for those children who reversed stunting (S2 Fig).

## Discussion

This study is the first to concurrently follow up post-SAM children and matched controls for an extended period after recovery from the index episode of SAM. It has shown that the post-SAM group remained at higher risk of AM and common childhood disease occurrence than controls for a period as long as 12 months after recovery and that catch-up growth does not occur, though a complete stunting reversal is observed in a small proportion of children

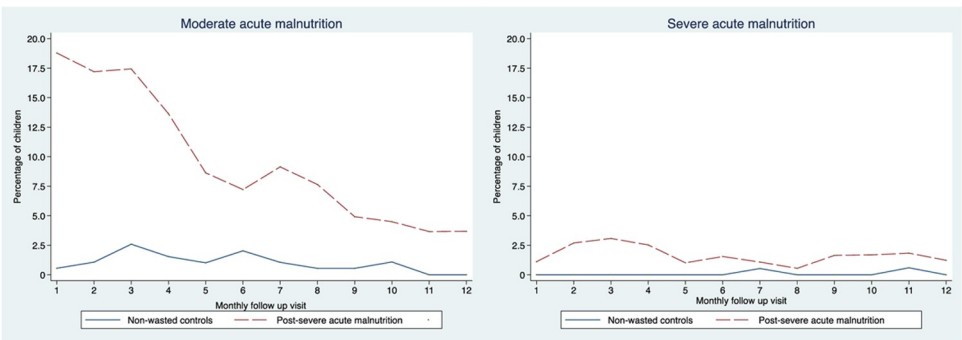

**Fig 2. Point prevalence (monthly) of moderate acute malnutrition (MUAC ≥11.0 cm and MUAC<12.0 cm) and severe acute malnutrition (MUAC<11.0 cm or bipedal oedema) by study group.**

**Table 2. Incidence rate of acute malnutrition in children post-severe acute malnutrition (n = 202) and control group (n = 201).**

|  | No. of episodes | Total person-time[a] | Incidence rate[a] | Incidence rate ratio (95% CI) | p-value[b] |
|---|---|---|---|---|---|
| Acute malnutrition[c] |  |  |  |  |  |
| Post-SAM[d] | 48 | 1297 | 3.7 | 5.5 (4.7, 15.9) | <0.001 |
| Control[e] | 14 | 2105 | 0.7 | 1.0 |  |
| Moderate acute malnutrition[c] |  |  |  |  |  |
| Post-SAM[d] | 34 | 1345 | 2.5 | 4.1 (2.1, 8.4) | <0.001 |
| Control[e] | 13 | 2105 | 0.6 | 1.0 |  |
| Severe acute malnutrition[f] |  |  |  |  |  |
| Post-SAM[d] | 26 | 2044 | 1.27 | 14.1 (3.5, 122.5) | <0.001 |
| Control[e] | 2 | 2216 | 0.09 | 1.0 |  |

[a]Expressed in 100 person-months

[b]Two-tailed exact mid-p test

[c]Children with a Mid-Upper Arm Circumference<12.0 cm at enrollment not included in the analysis

[d]Post-SAM = group for children enrolled at graduation from treatment of severe acute malnutrition (cases group)

[e]Controls = group for the non-wasted matched controls (control group)

[f]Children with Mid-Upper Arm Circumference<11.0 cm were not included in this analysis.

stunted at enrolment. These results can significantly contribute to the design of post-SAM interventions.

The recurrence of AM after recovery has been the focus of several publications recently [9, 16, 27]. Thus, there is now sufficient evidence to suggest that post-SAM children need special attention, including sustained follow-up and correction of persisting deficits even when they have recovered based on anthropometry [13, 28, 29]. This study has shown that this special attention is required throughout the 12 months after recovery as vulnerability to AM and infection persisted throughout the 12 months follow-up period. Other studies have shown a shorter period of vulnerability, indicating that in some contexts, a short 3-month period of

**Table 3. Incidence rate for common morbidities among children of post-severe acute malnutrition group and among the matched controls group.**

| Symptoms |  | N | No. of episodes | Follow-up time [a] | Incidence rate[a] | Incidence rate ratio (95% CI) | P value |
|---|---|---|---|---|---|---|---|
| Any illness |  |  |  |  |  | 1.39 (1.14–1.71) | <0.001 |
|  | Post SAM | 200 | 200 | 982 | 20.37 |  |  |
|  | Control | 199 | 198 | 1356 | 14.60 |  |  |
| Diarrhoea |  |  |  |  |  | 1.77 (1.37–2.29) | <0.001 |
|  | Post SAM | 198 | 146 | 1221 | 11.9 |  |  |
|  | Control | 198 | 110 | 1631 | 6,7 |  |  |
| Fever |  |  |  |  |  | 1.71 (1.33–2.20) | <0.001 |
|  | Post SAM | 200 | 151 | 1216 | 12.42 |  |  |
|  | Control | 198 | 114 | 1573 | 7.25 |  |  |
| Cough |  |  |  |  |  | 1.79 (1.39–2.30) | <0.001 |
|  | Post SAM | 199 | 155 | 1248 | 12.42 |  |  |
|  | Control | 198 | 111 | 1600 | 6.94 |  |  |
| Difficulty of breathing |  |  |  |  |  | 1.71 (1.28–2.29) | <0.001 |
|  | Post SAM | 196 | 118 | 1519 | 7.77 |  |  |
|  | Control | 198 | 81 | 1781 | 4.55 |  |  |

[a]Expressed in 100 person-months. Any illness includes either of diarrhea, fever, cough or difficulty of breathing.

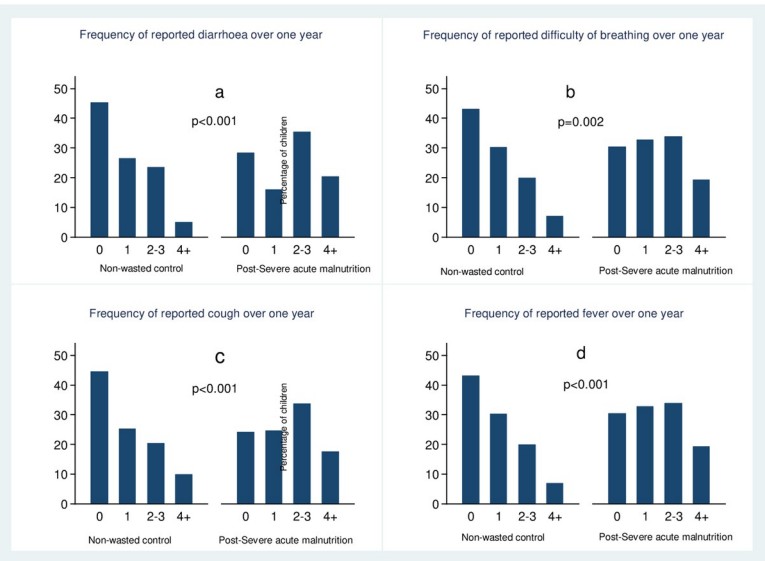

**Fig 3. Distribution of number of reported morbidity per child over one year by study group.**

post-discharge care may be sufficient [9, 16]. However, in addition to the timing of AM recurrence, other parameters should be considered for the decision on the optimal period of post-discharge follow-up, including immune system reconstitution, correction of anaemia, and other micronutrient deficiencies. Whilst some studies have shown that by the time children reach anthropometric discharge criteria, most nutritional and immune system (measured by thymus size) markers have returned to normal, other studies have shown that deficiencies remain [12, 13, 28–31].

In addition to defining the duration of post-discharge follow-up, there is also a need to define the package of interventions to support sustained recovery and restore normalcy of all physiological functions. Currently, evidence is lacking as recently tested post-discharge interventions, including infection control, food supplementation, micronutrient supplementation, and a combination of these approaches, have yielded inconclusive results [32, 33]. However, assuming current research efforts are sustained, including testing new interventions such as those to address maternal mental disorders or environmental enteric dysfunction, there are chances that a package of post-treatment support to prevent the recurrence of acute malnutrition will be available shortly [34–36].

If post-recovery follow-up were to be introduced in the management of SAM, potential options include either integrating follow-up into a routine Growth Monitoring and Promotion (GMP) programme or adding a follow-up phase to the CMAM protocol. Our results advocate for the former, as such integration will allow a longer surveillance of these post-SAM children. Indeed, it may be challenging to manage the extra workload that an additional CMAM follow-up phase would require in many high-burden countries.

Correction of body mass deficit is among the primary objectives of SAM treatment [10, 37, 38]. Based on the comparison of post-SAM and controls we can conclude that this objective was only partially reached. Whilst most post-SAM children met the treatment discharge criteria recommended at the time and grew at the same speed as the controls, the body mass of these children remained smaller than that of matched controls. Moreover, no catch-up trend was observed during the subsequent 12 months. However, this conclusion is valid only if Post-SAM had a growth trajectory similar to that of controls prior to developing the episode of

**Table 4. Evolution of anthropometric parameters according to the study group.**

| | Post-SAM[a] mean±SD[e] (n = 163) | Controls[b] mean±SD[e] (n = 167) | Difference[c] (95%CI)[f] | p-value[d] |
|---|---|---|---|---|
| **Weight, kg** | | | | |
| Baseline | 8.0±2.0 | 9.7±2.1 | 1.7 (1.2; 2.1) | <0.001 |
| Month 12[g] | 10.0± 2.1 | 11.5 ± 2.1 | 1.5 (1.1; 2.0) | <0.001 |
| Difference[h] (95% CI) | 2.0 (1.; 2.2) | 1.8 (1.7; 2.0) | -0.2 (-0.4;0.0) | 0.106 |
| p-value[9] | <0.001 | <0.001 | | |
| **Height, cm** | | | | |
| Baseline | 72.1 ±7.6 | 78.0 ± 9.6 | 5.9 (4.0; 7.9) | <0.001 |
| Month 12[g] | 80.1 ± 7.5 | 85.4 ± 9.0 | 5.3 (3.5; 7.1) | <0.001 |
| Difference[h] (95% CI) | 8.0 (7.3; 8.6) | 7.4 (6.8; 8.0) | -0.6 (-1.5;0.3) | 0.207 |
| p-value[i] | <0.001 | <0.001 | | |
| **Mid-upper arm circumference, cm** | | | | |
| Baseline | 12.7± 1.1 | 13.7± 0.9 | 1.0 (0.8; 1.2) | <0.001 |
| Month 12[g] | 13.7±1.3 | 14.6±1.3 | 0.8 (0.6; 1.1) | <0.001 |
| Difference[h] (95% CI) | 1.0 (0.8; 1.2) | 0.8 (0.6; 1.0) | -0.2 (-0.5;0.1) | 0.199 |
| p-value[i] | <0.001 | <0.001 | | |
| **Weight-for-height Z-score** | | | | |
| Baseline | -1.2±1.3 | -0.5±1.2 | 0.7 (0.5; 1.0) | <0.001 |
| Month 12[g] | -0.7±1.7 | -0.2±1.4 | 0.5 (0.1; 0.8) | 0.003 |
| Difference[h] (95% CI) | 0.5 (0.2; 0.7) | 0.3 (0.1; 0.5) | -0.2 (-0.5;0.1) | 0.125 |
| p-value[i] | <0.001 | 0.003 | | |
| **Height-for-age Z-score** | | | | |
| Baseline | -3.2±1.5 | -1.3±1.4 | 1.8 (1.5; 2.2) | <0.001 |
| Month 12[g] | -3.4±1.6 | -2.0±1.4 | 1. 4 (1.1; 1.7) | <0.001 |
| Difference[h] (95% CI) | -0.2 (-0.5; -0.0) | -0.7 (-0.9; -0.5) | -0.5(-0.8; -0.1) | 0.004 |
| p-value[i] | 0.043 | <0.001 | | |
| **Weight-for-age Z-score** | | | | |
| Baseline | -2.6±1.0 | -1.0±1.1 | 1.6 (1.3; 1.8) | <0.001 |
| Month 12[g] | -2.4±1.2 | -1.2±1.0 | 1.2 (0.9; 1.4) | <0.001 |
| Difference[h] (95% CI) | 0.2 (0.0; 0.4) | -0.2 (-0.3; -0.1) | -0.4(-0.6; -0.2) | <0.001 |
| p-value[i] | 0.011 | 0.004 | | |
| Height-for-age difference, cm | | | | |
| Baseline | -9.6±5.7 | -4.1±4.5 | 5.5 (4.4; 6.7) | <0.001 |
| Month 12[g] | -12.1±6.1 | -7.2±5.3 | 5.0 (3.7; 6.2) | <0.001 |
| Difference[h] (95% CI) | -2.5 (-3.1; -1.8) | -3.1 (-3.7; -2.4) | -0.6 (-1.5;0.3) | 0.215 |
| p-value[i] | <0.001 | <0.001 | | |
| Weight-for-age difference, kg | | | | |
| Baseline | -2.9±1.3 | -1.3±1.4 | 1.6 (1.3; 1.9) | <0.001 |
| Month 12[g] | -3.3±1.6 | -1.9±1.5 | 1.4 (1.1; 1.8) | <0.001 |
| Difference[h] (95% CI) | -0.4 (-0.6; -0.2) | -0.5 (-0.7;-0.4) | -0.1 (-0.3;0.1) | 0.331 |
| p-value[i] | <0.001 | <0.001 | | |

[a]Post-SAM = children who had recovered from severe acute malnutrition in community-based management of acute malnutrition program

[b]Controls = non-wasted matched community controls

[c]Between groups comparison

[d]unpaired t-test

[e]SD = standard deviation

[f]CI = confidence interval

[g] Only for children with 12[th] month follow up data

[h]Within group comparison of month 12 follow up and baseline parameters

[i]paired t-test.

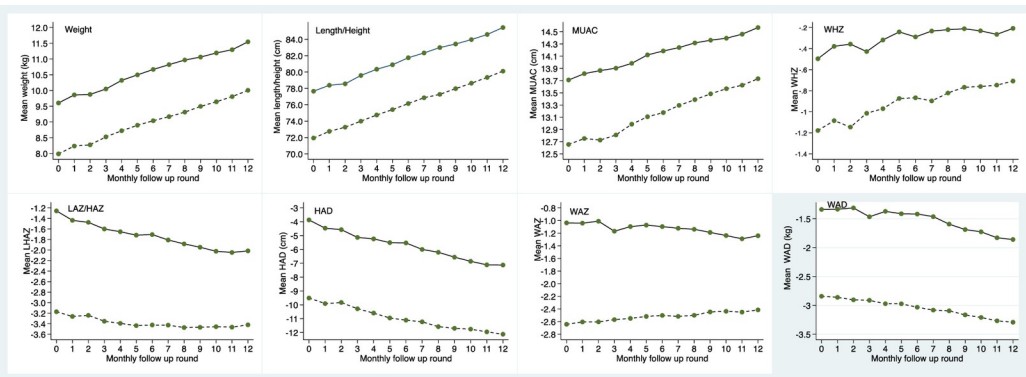

**Fig 4. Trend in the average anthropometric parameters by study group during the 12 months of follow-up.**

SAM for which they were admitted. There is a body of evidence that prenatal and early postnatal factors may lead to programmed growth restriction during childhood and even up to adulthood [39, 40]. Our results showing similarity of growth velocity between the Post-SAM and the controls during the follow-up period leads to the conclusion that they were growing well and suggests that the lack of catch up is unlikely to be due to factors operating during that period. The observed weight and length/height increments and the overall growth profiles allow us to formulate two complementary hypotheses. First, SAM disproportionally affects children by restricting growth along trajectories that follow the lower percentiles of the WHO 2006 growth curves. Second, by the time of graduation from the treatment programme, post-SAM had already corrected their growth deficits and subsequently grew as per their prenatally, and early infancy programmed growth trajectories. The first hypothesis is difficult to test as data on birth weight or growth prior to developing SAM were not available. However, it is worth mentioning that in both Post-SAM and non-wasted control groups, children who experienced an episode of SAM during follow-up were lighter and shorter at baseline than those who did not. The second hypothesis can be backed by prior literature suggesting differences in growth trajectories based on prenatal and early infancy influences [41, 42].

The hypotheses discussed above suggest that it might be unrealistic to expect that post-SAM children will ever fully catch up with the growth trajectory of non-wasted controls, especially when the deficit in length/height is large [43]. However, it is important to note that 14.4% of the Post-SAM group experienced a complete reversal of stunting without any specific intervention. This percentage could have been higher if these children had received an intervention to promote linear catch-up growth, leading to ponderal catch-up. However, evidence on interventions that can promote linear catch-up growth after malnutrition or growth restriction is limited. Indeed, research on the topic has so far yielded contrasting results making it difficult to propose a specific or a package of interventions for this purpose [43–52]. Thus, more research is needed to identify the best interventions for promoting growth catch-up following treatment of SAM.

Change in HAD is proposed as a better marker of linear catch-up than HAZ change [23, 53]. In our study, HAZ and HAD trends led to the same conclusion in the total sample and in those who had stunting reversal, suggesting that HAZ can be used to assess catch-up growth in the study setting. This result was observed despite inclusion of children above 24 months of age at baseline or those who crossed this age cut-off during the follow-up period. Thus, more studies are needed to demonstrate the benefit of HAD over HAZ in the evaluation of interventions' effectiveness on linear growth [53].

Our study had several strengths. First, the inclusion of comparison group that faced similar environmental and community contexts has allowed unbiased conclusions regarding the excess vulnerability of post-SAM children. The second strength was the prospective nature of the study that enhanced the accuracy of the collected data. Third, the monthly follow-up minimized the risk of missing short episodes of AM. The lack of data on pre-SAM and treatment period growth parameters is the major limitation of the study. Availability of these data could have allowed a better understanding of the post-discharge growth pattern. Both admission and discharge criteria of the Ethiopian government at the time were different from what most CMAM programmes currently use, and this could limit generalizability. Finally, we do not have biomarker data for more accurate assessment of nutrient status. As we proposed earlier, measurements including immune system reconstitution, correction of anaemia, and other micronutrient deficiencies would provide more reliable evidence of physiological and functional recovery.

In conclusion, this study has shown that post-SAM children remain at excess risk of AM and common disease than other children living in the same environment up to twelve months after graduating from treatment and that reoccurrence of SAM during this period should be considered differently than an episode diagnosed in a child without a history of SAM. Our results advocate for the design of post-discharge interventions that aim to prevent the reoccurrence of AM, reduce morbidity and promote catch-up growth. Further research is needed to improve the understanding of SAM episodes and SAM's effect on post-discharge growth patterns and to define the appropriate package of post-discharge interventions.

## Supporting information

**S1 Fig. Average height-for-age Z-score and height-for-age difference evolution of the different study groups over the follow period.**
(DOCX)

**S2 Fig. Average height-for-age Z-score and height-for-age difference evolution of children of the different study groups who had stunting reversal by the end of the follow period.**
(DOCX)

**S1 Table. Selected household characteristics of the participants by study group.**
(DOCX)

**S1 Dataset. Minimal anonymized data.**
(DTA)

## Acknowledgments

We thank the mothers and children who consented to give their time and support freely throughout this study. We are grateful for the hard work of the field data collectors, supervisors and the data entry clerks for their essential contribution. We thank the ENGINE, Jimma University and Valid International management teams for their vast contribution to the supervision: Beyene Wondafrash, Habtamu Fekadu, Erika Lutz, Cherinet Abuye, Basia Benda and Anne Walsh.

## Author Contributions

**Conceptualization:** Tsinuel Girma, Philip T. James, Hanqi Luo, Kate Sadler, Paluku Bahwere.

**Data curation:** Tsinuel Girma, Philip T. James, Hanqi Luo, Yesufe Getu, Yilak Fantaye, Paluku Bahwere.

**Formal analysis:** Tsinuel Girma, Alemseged Abdissa, Yesufe Getu, Paluku Bahwere.

**Funding acquisition:** Kate Sadler.

**Investigation:** Tsinuel Girma, Philip T. James, Alemseged Abdissa, Hanqi Luo, Yesufe Getu, Paluku Bahwere.

**Methodology:** Tsinuel Girma, Philip T. James, Alemseged Abdissa, Hanqi Luo, Yesufe Getu, Paluku Bahwere.

**Project administration:** Tsinuel Girma, Philip T. James, Alemseged Abdissa, Hanqi Luo, Yesufe Getu, Yilak Fantaye, Kate Sadler, Paluku Bahwere.

**Resources:** Tsinuel Girma.

**Supervision:** Tsinuel Girma, Alemseged Abdissa, Hanqi Luo, Yesufe Getu, Yilak Fantaye, Kate Sadler, Paluku Bahwere.

**Writing – original draft:** Tsinuel Girma, Paluku Bahwere.

**Writing – review & editing:** Tsinuel Girma, Philip T. James, Alemseged Abdissa, Hanqi Luo, Yesufe Getu, Yilak Fantaye, Kate Sadler, Paluku Bahwere.

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
