## [Decision Letter · Decision Letter 0]

5 Oct 2021

PONE-D-21-25723Nutrition status and morbidity of Ethiopian children after recovery from severe acute malnutrition: Prospective matched cohort studyPLOS ONE

Dear Dr. Girma,

Thank you for submitting your manuscript to PLOS ONE. After careful consideration, we feel that it has merit but does not fully meet PLOS ONE’s publication criteria as it currently stands. Therefore, we invite you to submit a revised version of the manuscript that addresses the points raised during the review process. Especially please provide clarification on the approach used to select controls, comparability of lost to follow-up and retained children and the applicability of the study findings to the existing SAM treatment protocol of the country. 

We look forward to receiving your revised manuscript.

Kind regards,

Samson Gebremedhin, PhD

Academic Editor

PLOS ONE

Journal Requirements:

Additional Editor Comments:

Abstract

Please provide key statistical figures with confidence interval or p-values including the reported 15 times higher risk of developing SAM.

Background

Please expand the last paragraph on extent of SAM relapse rate after discharge. How common is the problem?

Methods

Page 4, sample size estimation: “We projected an 8.5 difference…” per what population?How did you ascertain that the controls had no episode of acute malnutrition?How were the controls selected? I assume multiple age and sex matched controls would be available in the study villages.The matching criteria for age is vague. I assume matching based on age was made based on some tolerable rage. What age range did you tolerate?It is not clear how growth trajectory was defined or calculated.

Results

Please add a table that compares the basic socio-demographic and nutritional characteristics of lost-to-follow (including deaths and early exits) and retained subjects for both groups so that it can help readers to understand the likelihood of loss to follow up bias.

Table 1: I was expecting comparison based on basic socio-demographic variables including household wealth index or income and maternal educational status. Please include these variables in the list.

Figure 2: some of the contents in the boxes are not visible

Discussion

“Both admission and discharge criteria of the Ethiopian government at the time were different from what most CMAM programmes currently use, and this could limit generalizability”. This is an important point that requires clarification. What specific changes have been introduced to the protocol? How would that affect the generalizability of the study

Conclusion

What specific post-discharge interventions are you recommending for? Is that really financially and practically feasible to implement such longer supports to SAM cases? Is there any international experience before?

Reviewers' comments:

Reviewer's Responses to Questions

**Comments to the Author**

1. Is the manuscript technically sound, and do the data support the conclusions?

Reviewer #1: Yes

Reviewer #2: Yes

2. Has the statistical analysis been performed appropriately and rigorously? 

Reviewer #1: Yes

Reviewer #2: Yes

3. Have the authors made all data underlying the findings in their manuscript fully available?

Reviewer #1: Yes

Reviewer #2: No

4. Is the manuscript presented in an intelligible fashion and written in standard English?

Reviewer #1: Yes

Reviewer #2: Yes

5. Review Comments to the Author

Reviewer #1: This is a good manuscript with a good design, well done statistical analysis and an interesting topic.The paper is very well written and discussed

Comments:

1.How exactly were the controls carried out? It just says the selection criteria but how it was done in practice on the field? It would be better to say it here.

2.It will be better to use the term “non exposed” instead of control

3.I think another limitation of this study is the fact that this study is unable to separate the effects of SAM from the effects of the physical, social and other environments in which these subjects evolve, especially since there are differences in the socio-economic and even dietary status of the two groups, as well as the status of their mothers (MUAC), as can be seen in Table 1 of the supplementary file

Reviewer #2: The longer-term outcomes of children after treatment for severe acute malnutrition is an important topic. The paper is clearly written, helpfully highlighting that anthropometry is being used as a proxy of physiological and immune recovery, the lack of linear growth and common relapses recovery typically seen. I also applaud the use of height and weight deficit, properly addressing the problems with assessing z scores over time, and of use of survival models. have minor comments only.

Please clarify if the SAM cases were uncomplicated, give some more information on their duration of treatment, and if the data are available, their starting MUAC or WHZ. The reason why this may be important is that children who began just below the threshold for SAM may more easily recover and have less relapse than children who began far below the threshold for SAM.

Under Methods, for ‘We projected an 8.5 difference between post-SAM and controls in incidence of acute malnutrition’ – what does the 8.5 refer to? Is it an absolute percentage difference, relative percentage, ratio? If a percentage difference, please more clearly state the estimated percentage in the control group with 95% CI.

I am concerned by starting with an anticipated loss to follow up of 20% because this does not occur randomly. Children who become lost to follow up are usually more frequently ill, malnourished or have died. Please explain the rationale for 20%, efforts made to trace families and approach to dealing with the potential bias created. Fortunately, the actual figure was lower, but these issues remain.

Please give more details of how controls were identified and recruited. They are usually more difficult to correctly enrol than cases. Location is not mentioned in matching – this may be fine but needs an explanation of how location was dealt with when enrolling controls. For example, if within the same woreda then this may be considered frequency matching.

In results, I suggest rewording ‘As shown in Table 1, no difference was observed between post-SAM and control groups for matching criteria at baseline except that the post-SAM children were significantly lighter and shorter (p<0.001)’ because weight and height were not matching criteria (only age and sex were mentioned in methods).

Likewise, ‘The burden of common morbidities was higher (p<0.001) among post-SAM than controls (Fig 3)’ should be more precisely worded than just the word ‘burden’ indicating in the text whether this refers to a rate, the number of events, or the proportion of children experiencing morbidity events.

Out of interest, were the post-SAM children who experienced accelerated linear growth younger?

Given that the data were collected over time, it would be ideal to give the data on mortality and new episodes of malnutrition as rates, e.g. per 1000 child-years or 100 child-months.

In the discussion, I would argue that correction of body mass deficit is not the primary objective of SAM treatment. The objectives are to reduce risks of mortality and illness, and to promote neurodevelopment. Improving body mass is one component of helping achieve that objective, but not the only one. Others include improving body composition, correcting micronutrient deficiencies, identifying and treating medical conditions, identifying and addressing contributory home circumstances, including maternal mental health. Focus on body mass gain may be among the reasons we continue to see mortality, relapse, failure of height growth etc.

Might the apparent 14.4% reversal in stunting be due to regression to the mean or observation bias from loss to follow up?

6. PLOS authors have the option to publish the peer review history of their article (what does this mean?). If published, this will include your full peer review and any attached files.

Reviewer #1: **Yes: **Mwene-Batu Lyab Pacifique

Reviewer #2: **Yes: **James A Berkley

---

## [Author Response · Author response to Decision Letter 0]

27 Jan 2022

Response to reviewers and the revised documents are submitted as per the request from the Editor.

---

## [Editor Report · Decision Letter 1]

16 Feb 2022

Nutrition status and morbidity of Ethiopian children after recovery from severe acute malnutrition: Prospective matched cohort study

PONE-D-21-25723R1

Dear Dr. Girma,

We’re pleased to inform you that your manuscript has been judged scientifically suitable for publication and will be formally accepted for publication once it meets all outstanding technical requirements.

Kind regards,

Samson Gebremedhin, PhD

Academic Editor

PLOS ONE
---

## [Editor Report · Acceptance letter]

2 Mar 2022

PONE-D-21-25723R1 

Nutrition status and morbidity of Ethiopian children after recovery from severe acute malnutrition: Prospective matched cohort study 

Dear Dr. Girma:

I'm pleased to inform you that your manuscript has been deemed suitable for publication in PLOS ONE. Congratulations! Your manuscript is now with our production department. 

Kind regards, 

on behalf of

Dr. Samson Gebremedhin 

Academic Editor

PLOS ONE